# Evaluation of Aerobic Exercise Intensity in Patients with Coronary Artery Disease and Type 2 Diabetes Mellitus

**DOI:** 10.3390/jcm9092773

**Published:** 2020-08-27

**Authors:** Bernhard Schwaab, Mirca Windmöller, Inke R. König, Morten Schütt

**Affiliations:** 1Curschmann Klinik, Rehabilitation Clinic for Cardiology, Angiology and Diabetes, 23669 Timmendorfer Strand, Germany; mircawindmoeller@gmail.com; 2Institute of Medical Biometry and Statistics, University Hospital Schleswig Holstein, Campus Lübeck, University of Lübeck, 23562 Lübeck, Germany; inke.koenig@uni-luebeck.de; 3Institute for Diabetes and Endocrinology, 23552 Lübeck, Germany; morten.schuett@diabetes-luebeck.de

**Keywords:** aerobic exercise, anaerobic exercise, type 2 diabetes mellitus, coronary artery disease, glucose tolerance, exercise intensity

## Abstract

(1) Background: Physical activity is recommended in patients with type 2 diabetes mellitus (T2DM) and coronary artery disease (CAD) to reduce hyperglycemia and cardiovascular risk. Effective aerobic exercise intensity, however, is not well defined. (2) Methods: 60 consecutive patients performed cardiopulmonary exercise testing (CPX) of 30 min duration targeting a respiratory exchange ratio (RER) between 0.85 and 0.95, being strictly aerobic. Plasma glucose (PG) was measured before and after CPX as well as one and two h after exercise. Maximum exercise intensity was evaluated using a standard bicycle exercise test. (3) Results: 50 patients completed the protocol (62 ± 10 years, BMI (body mass index) 30.5 ± 4.9 kg/m^2^, HbA1c (glycated haemoglobin) 6.9 ± 0.8%, left ventricular ejection fraction 55 ± 8%). Aerobic exercise capacity averaged at 32 ± 21 Watt (range 4–76 Watt) representing 29.8% of the maximum exercise intensity reached. PG before and after CPX was 9.3 ± 2.2 and 7.6 ± 1.7 mmol/L, respectively (*p* < 0.0001). PG was further decreased significantly at one and two h after exercise to 7.5 ± 1.6 mmol/L and 6.0 ± 1.0 mmol/L, respectively (*p* < 0.0001 for both as compared to PG before CPX). (4) Conclusions: Aerobic exercise capacity is very low in patients with CAD and T2DM. Exercise at aerobic intensity allowed for significant reduction of plasma glucose. Individual and effective aerobic exercise prescription is possible by CPX.

## 1. Introduction

According to guidelines and position statements, patients with type 2 diabetes mellitus (T2DM) should be advised to be physically active in order to decrease both hyperglycemia and their cardiovascular risk [1,2,3]. In general, a moderate- to vigorous-intensity aerobic activity is recommended [1,2,3]. It is also proposed, however, that “greater exercise intensities tend to yield even greater benefits in HbA1c and aerobic capacity” [4] or that “higher levels of exercise intensity are associated with greater improvements in A1C and fitness” [1,5]. These recommendations are given for patients with T2DM without known coronary artery disease (CAD) [1,4,5].

In a pilot study, 10 patients with T2DM and concomitant CAD exhibited a very low aerobic exercise capacity of 29 ± 9 Watt (range between 10 and 45 Watt) as measured by cardiopulmonary exercise testing (CPX) [6]. Physical activity for 30 min with aerobic intensity reduced the two h plasma glucose (PG) in all patients substantially and average PG was reduced significantly as compared to PG before exercise. Exercise with anaerobic intensity, however, did not reduce PG in this small cohort [6]. Hence, it seems possible to detect a specific exercise intensity by CPX for effective blood glucose control in every single patient, being very different concerning their individual metabolic state [6]. In addition, the appropriate exercise intensity to control hyperglycemia might be different in patients with T2DM and CAD as compared to other cohorts with diabetes mellitus [7,8]. In this study, we evaluate the individual aerobic exercise capacity in a larger cohort of patients with T2DM admitted to cardiac rehabilitation (CR) with CAD. In addition, we measure the impact of aerobic exercise on PG until two h after cessation of physical activity.

## 2. Materials and Methods

Patients referred to CR after a recent acute coronary syndrome (ACS) or with stable CAD were included. In all patients, CAD was diagnosed by coronary angiography and T2DM was diagnosed using standard criteria [9]. During CR, oral glucose tolerance test was performed at the earliest on day 5 after ACS [10]. Exclusion criteria were insulin therapy, surgery within three months, chronic obstructive pulmonary disease, peripheral arterial disease, medication with cortisone, diseases of the exocrine pancreas, and infections. All patients gave their written informed consent before entering the protocol. The study was conducted in accordance with the Declaration of Helsinki and was approved by the ethics committee of the University of Lübeck (No. 15-019).

CPX was performed on an electronically braked, stationary cycle ergometer (Ergoselect 100, Ergoline GmbH, Bitz, Germany) at constant pedal speed of 50–60 rpm. Airflow, volumes and the O_2_, CO_2_ analysers were calibrated before each exercise session. Gases were analysed breath by breath (Cardiovit CS-200, Schiller Medizintechnik GmbH, Ottobrunn, Germany). After adaptation to the mask, parameters at rest were determined followed by a period of two min of unloaded cycling [11]. A 12-lead ECG was recorded continuously and blood pressure was measured at two-min intervals.

CPX was started with the lowest workload possible at 4 Watt and constant workload increments were tailored to the individual patient [11]. The automatic workload increase was stopped when the respiratory exchange ratio (RER) reached 0.85. At this point, the workload was titrated manually in steps of ±1 Watt to keep the RER between 0.85 and 0.95 over a time period of 30 min of cycling in order to perform a steady state aerobic exercise. After cessation of pedalling, RER must not exceed 1.00 during the recovery phase, demonstrating the definite aerobic intensity of this physical activity. CPX was terminated when O_2_ and CO_2_ had returned to resting values [11].

Exercise testing and training was performed at 9 a.m. in all patients. Patients were advised to have their breakfast and to take their oral anti-diabetic medication in the morning as they usually do at home, mimicking a realistic scenario of physical activity in a homelike setting. Plasma glucose (PG) was measured before CPX and immediately after the recovery phase. After CPX, patients were advised to remain seated, not to eat or smoke and only drink water. PG was measured again one and two h after CPX. Post-exercise PG was measured to detect the acute changes caused by physical activity. Within three days after CPX, a graded exercise test to exhaustion was performed to evaluate maximum exercise intensity [Watt] using the same stationary cycle ergometer. Workload was increased in two min stages and standard criteria for the cessation of the exercise test were used. At the beginning of CR, left ventricular ejection fraction (LVEF) [%] was measured by echocardiography using standard methods.

## 3. Statistical Analysis

Data in the entire group as well as in relevant subgroups are given as described by mean values ± standard deviations as well as median and range. Comparisons between plasma glucose values before and after exercise, after one and after two h were made using Wilcoxon tests reporting two-sided *p* values. The relationship between aerobic exercise capacity [Watt], maximum exercise intensity [Watt] and left ventricular ejection fraction [%] was analyzed using Spearman correlations with two-sided *p* values and described by linear regression models. To account for the multiple testing of four hypotheses, the global significance level of 0.05 was corrected, so that *p* values < 0.05/5 are considered statistically significant. Subgroup analysis was not performed due to small numbers.

## 4. Results

Inclusion criteria were met by 60 consecutive patients. CPX had to be discontinued prematurely in 10 patients (16.7%) due to hyperventilation (*n* = 4) and hypertension (*n* = 3). In two patients, it was not possible to hold RER constant between 0.85 and 0.95 and one patient became anaerobic with an RER > 1.00 despite maximum reduction of exercise intensity. These patients were excluded from statistical analysis. The study protocol could be completed in 50 patients, 10 female, 40 male (Table 1).

Primary cardiac diagnosis was ACS in 40 out of 50 patients (80%) and 10 patients exhibited stable CAD. Coronary intervention with stent (PCI) was performed in 45 of 50 patients (90%). All patients were medicated with platelet inhibition and five patients had additional oral anticoagulation. Patients received statins (86%), ezetimibe (13%), RAAS-Inhibition (92%) and beta blockers (86%), respectively. In 35 patients (70%) T2DM was known between 2 and 20 years (median 5 years) before participation in CR. In 15 patients T2DM was newly diagnosed during CR. Patients with known T2DM had pharmacological treatment, mostly by metformin (94%). Dual combination of anti-diabetic drugs was given to 28% of these patients and 10% received three antihyperglycemic drugs (SGLT2 inhibitors (*n* = 6) and GLP1 agonists (*n* = 3), each in combination with metformin). No patient was medicated with sulfonylurea. 14 out of 15 patients with newly diagnosed T2DM wanted to start with lifestyle interventions before beginning a pharmacological treatment.

Aerobic exercise (CPX) could be performed for 30 min duration without any problems in 22 of 50 patients (44%). In 28 patients (56%), CPX had to be terminated early due to muscular fatigue (*n* = 25), muscular pain (*n* = 1), burning feet (*n* = 1) and headache without hypertension (*n* = 1). Aerobic exercise capacity averaged at 31.8 ± 20.9 Watt (range 4–76 Watt). Further exercise parameters obtained during CPX are displayed in Table 2. Aerobic exercise capacity (Watt) was not correlated with LVEF (%) (r = −0.05; *p* = 0.71) but with maximum exercise intensity (Watt) (r = 0.39; *p* = 0.01; y = −0.101 + 0.297x; y = aerobic exercise capacity (Watt), x = maximum exercise intensity (Watt)).

In 49 out of 50 patients information about smoking could be retrieved retrospectively from our archives. Only eight patients (16%) had no history of smoking. In 41 patients (84%), smoking burden ranged between 2.5 and 50 packyears (median 30 packyears). 21 out of 41 patients (51%) stopped smoking on the occasion of this cardiac event, and 19 patients (46%) had stopped smoking earlier, between 1.5 and 40 years ago (median 13 years). Only one patient kept smoking. Aerobic exercise capacity was not influenced by any of these variables.

Plasma glucose values (PG) are displayed in Figure 1 and Table 3. Immediately after aerobic exercise, PG was reduced in 45 out 50 patients (90%). One hour after CPX, PG was reduced in 40 out of 50 patients (80%); two h after CPX, PG was reduced in 49 out 50 patients (98%). Patients in whom PG was not reduced one h after CPX exhibited a higher age (65.2 ± 11 vs. 60.7 ± 9 years), higher BMI (33.2 ± 7 vs. 29.8 ± 3 kg/m^2^), larger waist (123 ± 15 vs. 110.4 ± 10 cm) and a higher HbA1c (7.2 ± 0.8 vs. 6.8 ± 0.7%). In addition, they exhibited a lower aerobic exercise capacity (22 ± 16 vs. 34 ± 21 Watt) and a shorter aerobic exercise duration (22.7 ± 6 vs. 26.7 ± 4 min).

## 5. Discussion

Despite a IA (Class of Recommendation I, Level of Evidence A) recommendation in all guidelines and position statements [1,2,3], implementation of regular physical activity in daily living as well as its acceptance as an important treatment tool is poor in the view of patients with T2DM [7,12]. A potential reason for this mismatch might be the fact that the general recommendation of “moderate-intensity aerobic physical activity”, as mediated by guidelines and physicians, is too abstract and not easy to transfer into specific daily activity for the patient. Even in the literature, definition of “moderate-intensity aerobic physical activity” is very heterogeneous encompassing brisk walking [13], exercising at 40–59% of heart rate reserve or oxygen uptake reserve [4], at 55–69% of maximum heart rate [4], at 12–13 of Borg rating of perceived exertion, (scale ranging from 6 to 20 [4]), at 40–60% of maximum oxygen uptake [14] or at 50–70% of maximum heart rate [14]. In addition, general advice might not be appropriate for the individual patient.

Especially, the request to exercise at greater intensities [1,4,5] might lead to frustration in obese and deconditioned patients with the consequence of significantly reduced adherence as compared to exercise prescriptions with moderate intensity [7,15] and motivational factors as well as patient’s preference are crucial for long-term maintenance of regular physical activity [16]. In addition, the recommendation of higher intensities might apply to patients with T2DM to prevent cardiovascular diseases [1,4,5]. In patients with T2DM and already prevalent CAD, however, there is no specific recommendation about the level of adequate exercise intensity for effective control of hyperglycemia [8].

In this study, the individual aerobic exercise intensity was evaluated by CPX in a larger cohort of 50 consecutive patients with T2DM and CAD. Average aerobic exercise capacity was very low at 32 ± 21 Watt (range 4–76 Watt; see Table 2). As in our pilot study [6], two h PG was substantially reduced when patients exercised at this low aerobic intensity in 49 out of 50 patients (98%) and two h PG was significantly lower as compared to PG before exercise (see Table 3 and Figure 1). In 10 patients, PG was not reduced at one h after CPX. These individuals were older, (less muscular mass?) with higher BMI and larger waist (more visceral adipose tissue?) and a higher HbA1c (worse metabolic state?). In addition, these 10 patients exhibited a lower aerobic exercise capacity and a shorter aerobic exercise duration. There was no difference concerning sex and diabetes duration.

As aerobic exercise capacity was also not correlated with left ventricular function in this study [6], metabolic state, adipose tissue and muscular fatigue might be the determining factors for aerobic exercise capacity in patients with T2DM and CAD rather than cardiac disease in patients with mild to moderate reduced LVEF. Numbers were too small, however, to perform statistical testing in subgroups and the problem of heterogeneity could not be excluded. Maximum exercise intensity, as measured during a graded bicycle exercise test, was correlated with aerobic exercise capacity (y = −0.101 + 0.297x). As cardiopulmonary exercise testing needs more time and resources, it is not widespread in use [7], and it does not seem realistic to propose its application in all patients. Hence, it would facilitate clinical practice considerably if the individual aerobic exercise capacity could be predicted by a standard graded exercise test. Of course, this correlation has to be evaluated prospectively first.

There is one study evaluating a similar cohort of patients with T2DM and prevalent CAD or at high risk for cardiovascular disease [7]. This cohort (age 55 ± 9 years, mean duration of type 2 diabetes 7 years (median four years), weight 99.5 ± 22 kg, BMI 34.5 ± 7 kg/m^2^, HbA1c 7.7 ± 2%) was similar to the patients studied here (Table 1) and in our pilot study [6]. VO_2peak_ was also determined by CPX in all patients (*n* = 150) in the study of Jarvie et al. [7]. Again, cardiorespiratory fitness was “alarmingly low” in this specific cohort, exhibiting an average VO_2peak_ of 18.8 ± 5.0 mL/kg/min [7]. Bearing in mind that only 35% of the patients in this study had concomitant CAD as compared to 100% in our pilot study [6], VO_2peak_ was even further decreased in our earlier cohort, to an average of 15.9 ± 2.8 mL/min/kg [6]. These data show that cardiorespiratory fitness might be much lower in patients with T2DM and concomitant CAD [6] or at high risk for cardiovascular disease [7], as compared to other cohorts. As a consequence, recommendations for adequate physical activity to improve glycemic control and fitness might be different for specific patients depending on their metabolic state and their comorbidities.

In 1467 participants (55.6 ± 9 years, BMI 29.3 ± 6 kg/m^2^) with normal glucose metabolism (46%), impaired glucose tolerance (22%) and without insulin dependent diabetes mellitus (32%), the estimated energy expenditure (EEE) was assessed by structured interviews, and insulin sensitivity was measured [17]. EEE was defined as vigorous with approximately 25 min of an activity at a metabolic equivalent (MET) level of 7, and as non-vigorous with 50 min at a MET level of 3.5, and results were corrected for age, sex, dietary fat and alcohol intake, hypertension, smoking, ethnicity and clinical center. In patients with diabetes, vigorous EEE did not improve insulin sensitivity (Odds Ratio (OR (Odds Ratio)) 0.86; 95% CI −1.06 to + 2.82), whereas non-vigorous EEE did improve insulin sensitivity significantly (OR 2.64 (95% CI 0.11 to 2.64). In participants without diabetes, vigorous EEE increased insulin sensitivity as compared to non-vigorous EEE (OR 1.53 (95% CI 0.01 to 3.07) vs. 1.71 (−0.35 to + 3.81)) [17].

In 19 patients with metabolic syndrome and T2DM, moderate exercise intensity (*n* = 8, 52.0 ± 11 years, BMI 29.4 ± 5 kg/m^2^) and high exercise intensity (*n* = 11, 55.3 ± 13 years, BMI 29.8 ± 6 kg/m^2^) were compared in a randomized trial [18]. Moderate intensity was defined as exercising at 70% of the highest measured heart rate (HR_max_), and high intensity exercise was performed at 90% of HR_max_. After 16 weeks of training, fasting PG was significantly reduced by high exercise intensity (baseline 6.9 ± 0.6 vs. after 6.6 ± 0.6 mmol/L; *p* < 0.05) whereas moderate intensity did not reduce fasting PG (6.1 ± 0.5 vs. 6.5 ± 0.6). A positive two-h post load PG ≥ 11.1 mmol/L, however, was reduced from 7 out of 8 (87.5%) to 3 of 8 patients (37.5%) by moderate exercise intensity as compared with 8 of 11 (72.7%) to 4 of 11 patients (36.4%) by high intensity exercise training [18]. Considering that patients in this trial were extremely fit (VO_2peak_ 34 to 36 mL/min/kg) as compared to the patients in our study (VO_2peak_ 15.9 mL/min/kg), an absolute reduction of the fasting PG of only 0.03 mmol/L and a lower normalization rate of the two-h post load PG by high intensity (36% points), as compared to moderate intensity exercise training (50% points), does not justify the general recommendation of greater exercise intensities to yield greater benefits in HbA1c in patients with T2DM.

In the 2001 meta-analysis on the effects of exercise on glycemic control, Boulé et al. did not report a relationship between exercise intensity and improvement of glucose metabolism [19]. A meta-analysis, published in 2003 by the same group [20], revealed that exercise intensity predicted post-intervention weighted mean difference in HbA1c (r = −0.91, *p* = 0.002) to a larger extent than did exercise volume (r = −0.46, *p* = 0.26). However, only one study included an unequivocally vigorous exercise intervention at 75% VO_2peak_. Most of the studies exercised the patients between 50 and 65% of VO_2peak_ which is within the range detected as optimal in our study (Table 2). Thus, Boulé et al. concluded that high-intensity exercise might prove difficult or even hazardous for many previously sedentary people with T2DM and, lacking individual patient data, that one study’s positive result would not be sufficient to advocate high-intensity aerobic exercise for all people with diabetes [20].

In a randomized trial, continuous exercise training with moderate- and high-intensity was compared in 50 male obese patients with T2DM for six months duration [21]. When exercise bouts were matched for total energy expenditure, no interaction was observed between exercise intensity and the decrease in HbA1c and no correlation was found between changes in VO_2peak_ and HbA1c levels [21]. Bearing in mind the very low aerobic exercise capacity in our pilot study [6] in this cohort and in view of the results in the literature [7,17,18,19,20,21], the general recommendation of greater exercise intensities to yield greater benefits in HbA1c seems to be questionable. It could be possible that increased sympathetic nerve activity and elevated stress hormones during high-intensity exercise neutralize the glucose-lowering effect of physical activity or even increase plasma glucose [8].

## 6. Limitations

In 17% of patients, the method used was not applicable. Hence, CPX is not suitable for detecting the individual aerobic exercise intensity in all patients and measuring lactate might be an alternative in these individuals. We did not use the Borg scale to measure perceived exertion. Thus, our results cannot be compared to other studies using this valuable tool. Even though the number of patients studied (*n* = 50) is within the range of other studies [18,21], it still precludes generalization of our results and statistical analysis in subgroups could not be performed. The fact, that we did not adjust post-exercise glucose for the different durations of exercise might further confine generalization.

## 7. Conclusions

Optimal exercise intensity for regular physical activity is not well defined in patients with T2DM. Effective intensity might even be different according to whether exercise is prescribed as an antihypertensive agent, to control hyperglycemia, to reduce body-weight or to increase physical fitness. In addition, exercise intensity should be tailored to the individual patient as high-intensity physical activity is able to prevent metabolic syndrome in healthy elderly people [22] and to reduce fasting glucose in younger and very fit patients with T2DM [18]. As to the best of our knowledge, aerobic exercise intensity has not been evaluated in patients with T2DM and concomitant CAD. Aerobic exercise capacity in this cohort seems to be much lower than anticipated so far. Thus, exercise intensities, recommended for this cohort in guidelines so far might be too intense and might not reduce post-exercise glucose (as expected by patients and their doctors). However, our results demonstrate that low- to at most moderate-intensity aerobic exercise significantly reduced hyperglycemia in this deconditioned cohort. Individual aerobic exercise intensity can be determined by CPX, independently of metabolic state, cardiac function and physical fitness. We introduce an equation that might help to identify individual aerobic exercise intensity using a normal exercise test, available everywhere, instead of performing laborious and expensive cardiopulmonary exercise testing. As well as the evaluation of short-term effects, the long-term response of different exercise intensities should be investigated in large randomized trials [7,8], including personal preferences for a specific type of exercise in order to increase the adherence to sustainable lifestyle changes in patients with diabetes mellitus [15,16].

## Figures and Tables

**Figure 1 jcm-09-02773-f001:**
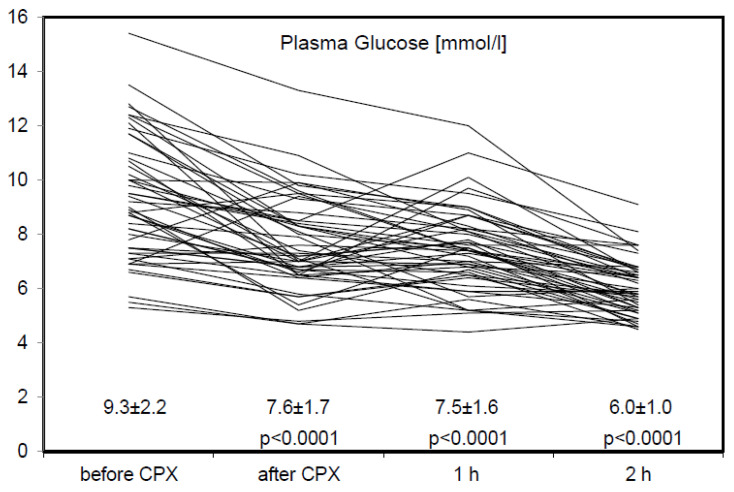
Plasma glucose during aerobic exercise (CPX). Plasma glucose (PG) is displayed in mmol/L as individual values for every patient (*n* = 50), measured before and immediately after 30 min of aerobic exercise (CPX) as well as 1 and 2 h after CPX. Statistical analysis is performed in comparison with PG before CPX.

**Table 1 jcm-09-02773-t001:** Basic patient data.

*n* = 50	Mean ± SD	Median	Range
age [years]	61.6 ± 9.6	62.0	34–80
diabetes duration [years]	4.6 ± 4.9	4.0	0–20
height [cm]	176 ± 0.1	175	158–193
weight [kg]	94.5 ± 16.4	91.0	62.0–136.0
BMI [kg/m^2^]	30.54 ± 4.86	29.54	21.97–44.41
waist [cm]	112.4 ± 13.6	110.5	90–147
HbA1c [%]	6.9 ± 0.75	7.0	5.8–9.3
LVEF [%]	55.6 ± 7.9	57.0	25–72

*n* = number of patients, BMI = body mass index, waist = circumference of the waist. LVEF = left ventricular ejection fraction, mean ± SD = mean value ± 1 standard deviation.

**Table 2 jcm-09-02773-t002:** Parameters during aerobic exercise cardiopulmonary exercise testing (CPX).

*n* = 50	Mean ± SD	Median	Range
aerobic exercise duration [min]	25.9 ± 4.8	28.0	15–20
aerobic exercise capacity [Watt]	31.80 ± 20.93	28.5	4–76
heart rate [/min]	86.24 ± 14.06	84	63–122
systolic blood pressure [mmHg]	137.6 ± 23.9	141	90–184
diastolic blood pressure [mmHg]	76.3 ± 10.4	77	51–104

CPX = 30 min of aerobic exercise, *n* = number of patients, mean ± SD = mean value ± 1 standard deviation, min = minutes.

**Table 3 jcm-09-02773-t003:** Plasma glucose before and after aerobic exercise (CPX).

Plasma Glucose [mmol/L]	Mean ± SD	Median	Range	*p*-Value
before CPX	9.3 ± 2.2	9.0	5.5–15.4	
after CPX	7.6 ± 1.7	7.2	4.7–13.3	*p* = 7.912 × 10^−5^
1 h after CPX	7.5 ± 1.6	7.4	4.4–12.0	*p* = 2.197 × 10^−5^
2 h after CPX	6.0 ± 1.0	5.9	4.5–9.1	*p* = 3.500 × 10^−15^

CPX = 30 min of aerobic exercise, h = hour, mean ± SD = mean value ± 1 standard deviation.

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
