# Peer review of "Evaluation of Aerobic Exercise Intensity in Patients with Coronary Artery Disease and Type 2 Diabetes Mellitus"

_jcm, 2020, doi:10.3390/jcm9092773_

Round 1

Reviewer 1 Report

The authors report results of a prospective, observational study that assessed the impact of aerobic exercise training on post-exercise glucose metabolism in persons with diabetes and coronary artery disease. Several larger studies have addressed these issues, but not generally in persons with coronary artery disease, as in this study. Furthermore, questions still remain about the most effective exercise "dose" for longer term glucose control in persons with diabetes.

A few concerns/questions:

  1. Did the authors account for the timing of each person's diabetes medications and the timing of the exercise training/testing? 
  2. The duration of exercise training differed in many of the patients, especially the 56% of patients who could not complete the 30 minutes exercise training regimen that was part of the study protocol. Did the authors adjust for this limitation?
  3. The authors focus on post-exercise glucose levels. Which is the most important measure of glucose control, post-exercise glucose level, fasting glucose, or hemoglobin A1C? Why did the authors choose post-exercise as their focus?
  4. The use of exercise testing to individualize exercise prescription is something that has been done for some time. Can the authors more fully explain the novel contributions of their study to the clinical application of exercise training?

Author Response

Article jcm-894896

Evaluation of Aerobic Exercise Intensity in Patients with Coronary Artery Disease and Type 2 Diabetes Mellitus

Bernhard Schwaab, Mirca Windmöller, Inke König, Morten Schütt

Point-by-point response to the reviewer No. 1

Ad 1.) We did account for the timing of each person`s diabetes medication and the timing of the exercise training/testing. Exercise testing and training was performed at 9 o`clock in the morning in all patients. In addition, patients were advised to have breakfast and to take their oral antidiabetic medication in the morning, in the same way, as they usually do at home. Doing so, we wanted to have an as much as possible realistic scenario, as if the patients would perform physical activity in their normal lives.

This information was added to the “methods“ section on page 2, line 83.

Ad 2.) We did not adjust for the different durations of exercise.

This information was added to the “limitations“ section on page 7, line 250.

Ad 3.) We decided to measure glucose levels because we wanted to detect the acute changes in plasma glucose that are caused by aerobic exercise. Our study is not suitable to decide which is the most important measure of glucose Control in the diabetic patient. This question is still open to discussion in the literature.

We choose “post-exercise“, because physical activity is the corner-stone of          nonpharmacological therapy during cardiac rehabilitation. In addition, exercise is recommended by all guidelines to control blood glucose. Hence, we focused on “post-exercise“ glucose.

This information was added to the “methods“ section on page 2, line 87.

Ad 4.) The novel contribution of our study is:

We truly evaluated the aerobic exercise intensity in patients with type 2   diabetes and coronary artery disease for the first time. We could not find this in the literature.

We demonstrated that the aerobic exercise intensity in this cohort is very much lower than anticipated so far.

Thus, the exercise intensities, recommended for this cohort in guidelines, is      too intense and might not reduce post-exercise blood glucose. (as expected by patients and their doctors …)

We introduce an equation that might help to identify the ideal exercise intensity by a normal exercise test, that is available everywhere, instead of a laborious   and expensive cardiopulmonary exercise testing.

We have pointed out these contents more clearly in the “conclusion“ section     on page 7, beginning at line 258.

Reviewer 2 Report

This Manuscript ID jcm-894896 entitled: “Evaluation of Aerobic Exercise Intensity in Patients with Coronary Artery Disease and Type 2 Diabetes Mellitus” by Bernhard Schwaab et al. investigates the individual aerobic exercise capacity in a cohort of patients with T2DM admitted to cardiac  rehabilitation after a recent acute coronary syndrome (ACS) or with stable coronary artery disease (CAD). In addition, they measure the impact of aerobic exercise on plasma glucose  until 2 hours after cessation of physical activity.

There is need for a more precise definition of “moderate-intensity aerobic physical activity”, which is present in the guidelines for T2DM. Thus, the aim of the study is of interest. However, some major issues in the study design came to my attention that are to be addressed.

In methods: Left ventricular ejection fraction (LVEF) [%] was measured by echocardiography using standard methods. When was echocardiography performed? At the beginning of rehabilitation? At the end? After CPX? It should be clarified.

-Table 1. Sex of patients is not specified, were they all men? Or which was the % of women? Sex-related differences may be present and thus should be carefully investigated.

-Smoking should also be reported.

-Patients characteristics show high heterogeneity, which is of concern. The number of 50 patients seems not powerful (as Authors briefly mention) to highlight and prove the role of metabolic determinants (in particular obesity) for the aerobic exercise capacity and the absent glucose reduction in ten patients.

very interesting would also be to see whether   after a  rehabilitation conducted with personalized aerobic exercise intensity an improvement in plasma glucose reduction was seen in all patients.

-Line 108: All patients with long history of type 2 diabetes had pharmacological treatment of blood glucose, mostly by metformin (94%). How long was the history of T2DM? and for how long was metformin used? These parameters are important and should be reported and evaluated in relation to the CPX results.

-Line 109: Dual combination of anti-diabetic drugs were given to 28% of these patients and 10% received 3 antihyperglycemic drugs. It would be very interesting to known which drugs patients were receiving as the newest GLP-1 agonists and SGLT2 inhibitors have major cardiovascular beneficial effects, which may have impacted the present findings.

Line: 125-126: Patients in whom PG was not reduced 1 h after CPX exhibited relation to T2DM specific drugs? T2DM history? Sex-related differences?

Many information which is now present in the discussion would better set the stage if briefly presented in the introduction.

MINOR

Please introduce abbreviations at their first use and then use them consistently.

For instance, CAD is used abbreviated line 104 but is introduced at line 153. Similar for plasma glucose (PG) see line 158 but used already abbreviated before.

Line 156: CHD, for coronary heart disease is not explained. It would be better to use always the same abbreviations (i.e. CAD) to facilitate readers.

Author Response

Article

jcm-894896

Evaluation of Aerobic Exercise Intensity in Patients with Coronary Artery Disease and Type 2 Diabetes Mellitus

Bernhard Schwaab, Mirca Windmöller, Inke König, Morten Schütt

Point-by-point response to the reviewer No. 2

LVEF [%] was measured at the beginning of rehabilitation in all patients.

This information was added in the “Methods“ section on page 2, line 91.

In our cohort of 50 patients, 10 patients were women and 40 patients were men.

This information was added in the “results“ section on page 3, line 108.

Due to the small number of patients, subgroup analysis, such as men vs. women, young vs. old, obese vs. non-obese could not be performed.

This information was added in the “statistical analysis“ section on page 3, line 101 and has been in the original manuscript in the “Limitations“ section on page 7, now on line 250.

Smoking habits were not prospectively assessed in this study. If the reviewer insists on this information, we could retrieve it manually from our paper folders in the archive.

The reviewer is right, heterogeneity is a problem in our data analysis. As mentioned before, due to the small number of patients, subgroup analysis, was not performed. In the “Discussion” on page 5, line 174, these differences, including BMI, are given without statistical analysis. We have added the problem of heterogeneity in the “Discussion” on page 5, line 179.

Unfortunately, we have no follow-up. We agree with the reviewer that it would be extremely interesting to see, whether our exercise prescription would results in a reduction of plasma glucose in all patients at the end of cardiac rehabilitation. Therefore, we stated in the “Conclusions“ on page 7, line 268: "Besides the evaluation of short-term effects, the long-term response of different exercise intensities should be investigated".

Line 108: In those patients with known diabetes mellitus, diabetes duration was between 2 years and 20 years with a median of 5 years. This information is added to the “Results“ section on page 3, line 116. We do not know, how long patients were medicated with metformin or other anti-diabetic drugs. Subgroup analysis concerning the medication was not performed.

Line 109: Patients with two or three anti-diabetic drugs received SGLT2 inhibitors (n=6) and GLP1 agonists (n=3) in combination with metformin. This information is added to the “Results“ section on page 3, line 120. Subgroup analysis concerning this medication was not performed.

Line 125-126: The number of patients in whom PG was not reduced immediately after exercise (n=5), 1 h after CPX (n=10) and 2 h after exercise (n=1) was very small. Hence, we were not able to perform subgroup analysis. In the “Discussion” on page 5, line 174, the differences are mentioned without statistical analysis. As suggested by the reviewer, we added diabetes duration and sex. The numbers concerning different medications were too small to derive any conclusion.

We have transferred some information from the “Discussion” to the “Introduction”, from page 5, line 158 to page 2, line 47.

We thank the reviewer very much for reading our manuscript attentively. All abbreviations were checked, corrected and introduced at their first use. We apologize for this mistake.

Round 2

Reviewer 2 Report

Authors have addressed almost all my comments. I think it would be important to add also the information about smoking habits, duration and years since cessation and assess  correlations of these variables with the Exercise capacity.

MINOR

Line 45, page 2:check” in A1C and fitness”. Should be HbA1c?

Author Response

Article

jcm-894896

Evaluation of Aerobic Exercise Intensity in Patients with Coronary Artery Disease and Type 2 Diabetes Mellitus

Bernhard Schwaab, Mirca Windmöller, Inke König, Morten Schütt

Response to the reviewer No. 2; second revision

In 49 out of 50 patients informations about smoking could be retrieved retrospectively from our archives. Only 8 patients (16%) had no history of smoking at all. In 41 patients (84%), smoking burden ranged between 2.5 and 50 packyears (median 30 packyears). 21 out of 41 patients (51%) stopped smoking on the occasion of this cardiac event, 19 patients (46%) had stopped smoking earlier, between 1.5 and 40 years ago (median 13 years). Only one patient kept smoking. Aerobic exercise capacity was not influenced by any of these variables.

This information was added to the manuscript on line 131, page 3.

MINOR

On line 45, page 2, we cite literally the spelling used in publication No. 5, “A1c“. We marked the citation by using quotation marks. Here, the HbA1c is meant, but we would prefer to use “A1c“ as used in this publication.